# A Framework for Transparency in Precision Livestock Farming

**DOI:** 10.3390/ani13213358

**Published:** 2023-10-29

**Authors:** Kevin C. Elliott, Ian Werkheiser

**Affiliations:** 1Lyman Briggs College, Department of Fisheries and Wildlife, and Department of Philosophy, Michigan State University, East Lansing, MI 48825, USA; 2Department of Philosophy, University of Texas Rio Grande Valley, Edinburg, TX 78539, USA

**Keywords:** precision livestock farming, transparency, philosophy of science, responsible innovation, open science, community epistemic capacity, stakeholder engagement, epistemic justice

## Abstract

**Simple Summary:**

The emergence of precision livestock farming (PLF) raises important issues for many different social groups, including farmers, consumers, regulators, and the food industry. This paper explores how those who develop PLF systems can communicate more effectively with different groups about the technologies that they are creating. We suggest that developers reflect on four issues: (1) the different kinds of information that various groups might want to know; (2) the audiences that might care about these different kinds of information; (3) the major difficulties involved in providing the information; and (4) potential strategies for overcoming those difficulties.

**Abstract:**

As precision livestock farming (PLF) technologies emerge, it is important to consider their social and ethical dimensions. Reviews of PLF have highlighted the importance of considering ethical issues related to privacy, security, and welfare. However, little attention has been paid to ethical issues related to *transparency* regarding these technologies. This paper proposes a framework for developing responsible transparency in the context of PLF. It examines the kinds of information that could be ethically important to disclose about these technologies, the different audiences that might care about this information, the challenges involved in achieving transparency for these audiences, and some promising strategies for addressing these challenges. For example, with respect to the information to be disclosed, efforts to foster transparency could focus on: (1) information about the goals and priorities of those developing PLF systems; (2) details about how the systems operate; (3) information about implicit values that could be embedded in the systems; and/or (4) characteristics of the machine learning algorithms often incorporated into these systems. In many cases, this information is likely to be difficult to obtain or communicate meaningfully to relevant audiences (e.g., farmers, consumers, industry, and/or regulators). Some of the potential steps for addressing these challenges include fostering collaborations between the developers and users of PLF systems, developing techniques for identifying and disclosing important forms of information, and pursuing forms of PLF that can be responsibly employed with less transparency. Given the complexity of transparency and its ethical and practical importance, a framework for developing and evaluating transparency will be an important element of ongoing PLF research.

## 1. Background

Precision livestock farming (PLF) is an important developing suite of technologies [1,2,3,4]. The goal of PLF is “to manage individual animals through continuous real-time monitoring of health, welfare, production/reproduction, and environmental impact” [2]. It involves collecting information about the well-being of livestock through a range of sensors and analytic tools that can generate data through images, sounds, heart rate monitors, accelerometers, chemical analysis of waste, and a range of other tools. Some of these tools are currently in use, others are being used on a small scale as proof-of-concept, and many others are only in development [5,6,7]. The data from these systems are typically analyzed using algorithms that turn the low-level data into meaningful information that can guide the decision-making of farmers. PLF also has the potential to be operated via “closed loop” systems, whereby systems can self-correct based on the collected data without depending on human guidance [8,9,10], though these systems lead to concerns that will be discussed in a later section of this paper. These closed-loop systems can be facilitated by machine learning applied to large datasets of instrument data and outcomes, which can generate both insights into connections between variables that human farmers would not look for or notice as well as faster and more accurate predictions of outcomes for animals based on limited information [11,12].

PLF has the potential to generate a number of benefits for farmers, consumers, and farmed animals. Nevertheless, PLF also generates or exacerbates preexisting ethical and social issues that need to be addressed if it is to be implemented in a socially responsible fashion. While a fully comprehensive review of potential benefits and concerns is beyond the scope of this paper, some examples will be helpful for our later discussion of how to reap the benefits while mitigating some of the concerns. For example, an example of potential benefits is that as the number of animals on farms increases, it can be more difficult to ensure their welfare, but PLF can help farmers keep closer track of their livestock [1,13]. For animals who experience stress in the presence of humans, PLF could also ease their stress by allowing them to be monitored with less direct human interaction [14]. PLF could also ease the workload on farmers by creating automated systems that address potential problems without requiring human intervention [8,9,10]. PLF can also promote sustainability [15,16], such as by enabling farmers to feed their animals with more precision, thereby avoiding waste [17,18], thus increasing both environmental and economic sustainability for the farm. PLF can even benefit consumers by facilitating more careful tracking of animals through the supply chain, thereby providing greater transparency for consumers who want to know where animals have been and how they have been treated [3].

As an example of potential concerns, PLF could contribute to the general trend toward consolidating smaller farms into larger ones, thereby altering rural communities and eliminating agricultural jobs [14]. One might also worry that PLF could be used as a “cover” to argue that agricultural intensification is compatible with protecting animal welfare, whereas critics might contend that farm animals would actually be better off on smaller, more traditional farms [6]. Privacy is another important ethical issue raised by PLF [3]. Given all the data collected through these systems, it will be essential to develop policies governing the sharing of this data with outside parties. Finally, one might worry about the ways that PLF could change the relationships between farmers and their livestock by eliminating the direct connections that farmers currently have as they assess the well-being of their animals [14,19].

## 2. Transparency and PLF

Many ethical issues have begun to be discussed in the scholarly literature on PLF; however, the issue of transparency about PLF has received relatively little attention thus far. There has been some discussion about the potential for PLF, in conjunction with technologies like blockchain (the decentralized transaction and data management technology most known for its use in cryptocurrency [20]), to provide transparency for consumers about the paths that animals have taken through the production process [3], but this literature does not focus on the need for transparency about the nature of PLF technologies themselves. Although a few authors have begun to call for this kind of broader transparency about the values embedded in PLF systems [6], there has been little discussion about how best to achieve this form of transparency or about the particular challenges that arise in doing so. This is an important gap because transparency will be crucial for pursuing PLF in a responsible fashion. The open science movement has recently highlighted the important role that transparency plays in promoting reproducible science, accelerating advances, and fostering public engagement with scientific research [21,22,23]. In the context of PLF, transparency is especially important because there are a number of values at play in this area of research (e.g., animal welfare, profit, sustainability, rural development, and so on). These values can come into conflict, and they can be interpreted in different ways [14], so transparency is important to enable farmers and consumers to decide what kinds of PLF systems actually help them to achieve their goals.

While it seems clear that transparency is important in this context, transparency is not a simple concept. As one of the authors (Elliott) has put it: “In a very basic sense, something is transparent when one can see through it. Thus, transparency is used as a metaphor in fields like politics and science to express the notion that information or processes have been ‘made visible’” [24]. However, he goes on to point out that this basic idea of making information visible can involve a great deal of complexity in practice. He proposes a taxonomy that distinguishes different forms of transparency in terms of their purposes, the audience for the information being provided, the content of the information being provided, and the timeframe, actors, mechanisms, and venues through which the information is provided [25]. He also notes that there are dangers associated with pursuing transparency, and those dangers have to be weighed against the benefits.

Although it is not necessary for our purposes to go into all these dimensions, it is helpful to keep in mind some of the major reasons for pursuing transparency and concerns about doing so. One possible reason for pursuing transparency is the notion that it is either inherently good or that an essential feature of scientific practice is to be open about one’s work [26]. But even if one rejects this “intrinsic” argument for transparency, there are a wide variety of instrumental reasons for pursuing it. As intimated above, these include: (1) promoting higher-quality science by facilitating external scrutiny of it; (2) promoting faster innovation; (3) fostering trust; (4) fostering greater diversity and inclusion in the scientific community; and (5) equipping the recipients of information to make better decisions for themselves. These benefits need to be weighed against a variety of potential concerns: (1) using up limited resources, including time, in an effort to provide information; (2) generating confusion on the part of those receiving information; (3) revealing private information or confidential business information; (4) assisting “bad actors” who aim to use the information inappropriately; and (5) creating rigid requirements that detract from the diversity of the scientific community’s methods or practices [27]. Elliott emphasizes that one can typically address these concerns without abandoning the pursuit of transparency altogether [25]. One of the benefits of thinking about transparency in terms of a taxonomy with multiple dimensions is that one can explore different forms of transparency that are not as susceptible to concerns. For example, even if it would violate research participants’ privacy to provide detailed information about them, it might still be possible to provide some transparency in the form of more limited or de-identified information about the participants. Similarly, if it would threaten confidential business information to provide all the data underlying a study, it might still be possible to provide some transparency in the form of information about the methods and principles used to analyze the data.

This paper draws on these insights in order to propose a framework for pursuing transparency about PLF. Building on the more general taxonomy of transparency developed by Elliott [25], the framework consists of four parts: audience, content, challenges, and strategies (see Figure 1; Table 1). According to this framework, those seeking to promote transparency about PLF should first consider, in specific contexts, the audiences toward which they are striving to provide information. Building on this consideration of audiences, they can consider the specific content that is most relevant to disclose. In order to communicate this content in a meaningful way, it is important to recognize the challenges that can make it difficult to achieve. Finally, drawing on the other elements of the framework, strategies can be developed for achieving meaningful forms of transparency. The following section employs philosophical methods of analysis (especially conceptual clarification) to elaborate on each of the four elements of the figure in the context of PLF. It provides examples of major audiences, content, challenges, and strategies that could be important to consider in a PLF transparency initiative, with the understanding that further empirical investigation of each element would help provide additional guidance for those seeking to implement the framework. Although this framework has been developed specifically for application to PLF, many of its features could also be applicable to other areas of agriculture and biotechnology in general.

## 3. A Framework for Transparency

### 3.1. Audience

In order to provide appropriate transparency, it is important to consider the audience toward which information is being directed because different audiences have different informational needs and different ways of obtaining information. For example, the open science movement has generally focused on communicating information in ways that serve other scientists and technologists. While some elements of the open science movement can be helpful to non-specialists (e.g., publishing articles in open access formats), most features of the open science movement (e.g., making raw data available in publicly accessible databases) are geared primarily toward the scientific community. To meet the needs of non-specialists, it is often insufficient merely to make information available; the information generally needs to be interpreted in ways that are meaningful to them [21]. In the context of PLF, we suggest at least five different audiences that could have unique informational needs: scientists and engineers, farmers, consumers, industry groups, and regulators.

Scientists and engineers are likely to be interested in fairly traditional elements of open science, such as open data and open access to research materials [28]. In contrast, farmers are less likely to want this technical information and are more likely to want the “take-home” lessons about what these systems can do, how they work, and what their limitations are [5,29]. Consumers would probably not even care about the working of the systems, but at least some consumers might be interested in the “implicit values” associated with the systems (e.g., whether animal products from farms that employ PLF systems promote particular values concerning animal welfare or sustainability) [30]. Various industry groups, such as meat-packing companies, distributors, wholesalers, grocery stores, and restaurants, are likely to have a mixture of informational needs that could vary depending on how closely they work with farmers, regulators, or consumers. Finally, regulators are likely to be interested in the extent to which PLF systems can be designed to ensure compliance with regulations, whether they could inadvertently violate them, and whether compliance with regulations would become more or less difficult to verify when using the systems. Of course, these five categories do not include all the audiences that could be considered. For example, PLF system designers are often called on to consider “society” or the “public” [5]. However, we contend that this is such a large group with so many different informational needs that it cannot be usefully analyzed as a single audience in this model. Instead, it would need to be broken down into different interest groups. We also acknowledge that we have characterized audiences in terms of their likely informational needs, but it would be important to actually interact with these audiences in order to determine their informational needs in more concrete detail.

### 3.2. Content

The second component of our framework for transparency about PLF is the content to be disclosed. We have already seen that different audiences are likely to care about different kinds of information. Without providing an exhaustive discussion of all the kinds of content that could be discussed, this section probes more deeply into five major categories of information about PLF that could be disclosed as part of a transparency initiative. The first category of information concerns the basic workings of a PLF system, including its major strengths and limitations. For example, people might want to know what features of the animals the system measures, what outputs the system strives to maintain, the basic features of how the system functions, the conditions under which the system was developed and tested, and the safeguards that are in place to prevent the system from malfunctioning. When new PLF systems are being implemented, audiences will want to know enough about them to feel comfortable that they will work successfully. They may also want to have this information in comparison to other existing or possible alternative PLF systems.

A second category of information focuses not so much on how PLF systems work but on the data generated by them. For example, consumers and regulators might want to receive information about how often animals experience disease or other forms of stress. They might also want to know how often closed-loop PLF systems need to make particular sorts of adjustments and what those adjustments are. In addition, they might want to use the information generated by PLF systems to track the movements of animals through the agricultural system or to ensure that the animals meet criteria for particular kinds of certification (e.g., organic or cage-free). All the groups mentioned in the previous section may well also be interested in comparing data from one operation to others (though in this case some of those groups, most notably farmers and industry groups, would both want that information and possibly want to keep private information about their own operation).

In addition to the basic information about how a PLF system works and the data generated by it, a third major form of content concerns the goals that the system was designed to promote. For example, one might wonder whether the developers were particularly focused on efficiency, animal welfare, or environmental sustainability and how they prioritized those values when they came into conflict. One might also wonder how they defined those concepts, such as what elements of animal welfare or environmental sustainability they focused on. Research has shown that farmers, to pick one audience as an example, often have a few values they prioritize and want to actively maximize, but at the same time see other values as constraints on that maximization rather than something they are also trying to maximize [31].

A fourth category of content about PLF systems concerns the implicit values embedded in them. This is a more difficult form of information to disclose because it is not always obvious to those developing and working with the systems. Implicit values arise when developers make particular choices when designing systems (e.g., measuring particular things, analyzing the data in particular ways), and those choices end up serving some values rather than others (e.g., promoting animal welfare over profit, or vice versa) [32]. This form of content can be very significant because even if the users of PLF systems feel comfortable that their values are generally aligned with those of the system designers (e.g., desiring to promote some particular definition of animal welfare), they might still worry that the system could inadvertently reflect implicit values that they disagree with. For example, when testing the system, the developers might have regarded a 95% success rate at identifying injuries to the animals as adequate. However, some users might have higher standards and would have wanted something like a 99% success rate in order to feel comfortable relying on the PLF system. Along the same lines, developers might be concerned with the overall rate of injuries, while some users might be more concerned about the injuries to mothers, the young, the weak, etc. Developers of a PLF system have to make numerous decisions about how accurate all their sensors need to be, what endpoints they need to measure, and how to handle tradeoffs between optimizing different features of the system. All of these choices can make the system as a whole more prone to various sorts of errors, and the users of the system might not agree with the developers’ implicit values about what kinds of errors are most important to avoid and what frequency of errors is acceptable.

A fifth form of content about PLF systems is even more fraught with difficulties. This content concerns the operation of the machine learning algorithms associated with some PLF systems. Because the working of machine learning algorithms is typically not comprehensible to human beings, it raises particularly important issues related to transparency [33,34,35]. Some users of PLF systems might want to know, for example, how confident they should be that the algorithms will generate reliable conclusions in the context of their farms. They might also want to know whether the concerns about implicit values discussed in the previous paragraph might apply to the machine learning algorithms. For example, although most farmers probably would not be interested in the details about how these algorithms were developed and how they operate, they might have the general worry that the algorithms might not be prioritizing exactly the same features of animal welfare as the farmers. For instance, the algorithms might be designed to promote the animals’ growth or freedom from disease, whereas some farmers might be more concerned about the animals’ activities or subjective experiences. Users might also be concerned with the likelihood that the algorithms are finding specious connections between inputs and outcomes that a human would correctly judge as an artifact of limited or incorrect data. Without receiving more information about the nature of these machine learning algorithms, many users might feel uncomfortable making use of them in their farming operations [33].

### 3.3. Challenges

The next component of our proposed framework consists of the challenges associated with pursuing transparency. Many of these challenges have already emerged from our discussion of different audiences and content. For example, one challenge is that those developing PLF systems might not have a clear understanding of the different audiences they need to consider and the kinds of content they want to know. Even if they could identify the relevant audiences, they might have difficulty, in some cases, explaining the technical details of how their systems work. Especially in the case of closed-loop systems that gather information about the animals and make automatic corrections in response to the available information, the systems might be too complex to explain easily to those who might want to know how they operate.

But even if the developers could find a way to disclose all the detailed information that some audiences might want, another challenge is that the developers themselves might be ignorant of some relevant information. As discussed above, some audiences might want to know how PLF systems implicitly promote some values (e.g., particular conceptions of animal welfare) over others. However, when developing new technological innovations, it is often unclear—even to the developers—how they promote particular values or interests over others. One might think that this is not specifically a problem related to *transparency*; rather, it might seem to be merely a function of the complexity of PLF systems and the limited perspective associated with particular disciplinary approaches. However, it is important to keep in mind the full breadth of the conception of transparency associated with Elliott’s taxonomy [25] and the framework that we are developing here. In Elliott’s view, the audiences for transparency can include not only external stakeholders but also the scientists and engineers who are working on a project. Important values and assumptions associated with the project may not be clear to them, and one of the goals of a transparency initiative could be to help them develop a richer understanding of these project features so that they can in turn be more transparent with others outside the project.

Another challenge to successful communication is trust between developers and potential users. If those trying to communicate the information do not trust the receivers, it is possible that they will try to manipulate them to obtain a desired outcome or to protect themselves. The communicators might also simplify the information they are communicating to the point that it becomes unhelpful, misleading, or even incorrect. If the receivers do not trust the communicators, they may not be able to take up any of the information they are being provided, even if it is in their own interest to do so. While some trust issues have to do with presentation style and other tools of rhetoric, and some trust issues have to do with the creation and maintenance of relationships between the various groups, it is also the case that previous harmful incidents can lead to justifiably low trust in ways that are quite difficult to overcome [36].

A separate but related challenge to achieving transparency involves the motivations and interests of those offering or receiving the information. For example, those promoting PLF systems might prefer not to acknowledge some of the systems’ weaknesses or the ways the systems prioritize some values over others. In addition, those using the systems might not want to disclose certain kinds of data generated by the systems (e.g., about rates of disease or injuries or other animal welfare concerns). Although this reticence to share information might sometimes be narrow-minded and self-serving, it could also reflect the legitimate concern that those receiving the information could misinterpret it and draw illegitimate conclusions. It could also reflect companies’ concerns about protecting their intellectual property and safeguarding confidential business information. These IP and related concerns might differ between different countries’ IP regimes, making communication across national boundaries more difficult. For farmers, the fear of providing detailed information about their operations might even motivate some of them not to adopt PLF systems at all.

Finally, the use of machine learning algorithms in PLF systems raises special challenges. In some cases, the challenge might just be that the recipients of information about the algorithms do not have the background knowledge to understand them. Although this might not initially seem to be a failure of transparency (because information about the algorithms is available), we would classify it as a failure of transparency because the available information is not understandable or usable for the intended audience. An additional challenge is that in the case of machine learning algorithms, even the developers might not know what factors are responsible for the algorithms’ outputs. As a result, the developers might be unable to provide a number of other relevant pieces of information. For example, they might not be able to identify the precise conditions under which the systems could become unreliable. They also might be unable to identify important biases, limitations, or “blind spots” that affect the systems. These limitations could be caused by biases in the training data used to develop the system, or they could be a function of the particular phenomena that the algorithms focus on. Without understanding how the algorithms work, it could be very difficult to provide detailed information about their strengths and weaknesses and the implicit values associated with them. This is particularly the case for the subset of machine learning commonly referred to as “deep learning” algorithms, in which programmers do not set which aspects of the environment the system is tracking.

### 3.4. Strategies

The final component of our transparency framework is to explore strategies for providing relevant content to audiences in a manner that overcomes major challenges. Although different situations and challenges are likely to call for different strategies, some general ones could prove helpful under a variety of circumstances. For example, one important kind of strategy is for the developers of PLF systems to collaborate with end users during the development process. One benefit of this co-creation process is that it helps the end users understand the major features of PLF systems, and thus it provides a form of transparency that would be difficult to provide otherwise. In addition, when users and developers collaborate, they are more likely to identify ways in which implicit values could be embedded in the operation of the systems. Thus, this process of collaboration “early and often” [37] can be especially helpful for uncovering features of PLF systems that could be important to disclose but that would not have even been recognized otherwise. (See Thompson et al.’s discussion of an “ethical matrix” for a useful framework for these stakeholders to express their ethical concerns [6]).

Another, and somewhat less ambitious, kind of strategy to address transparency challenges is to design PLF systems in ways that require less transparency. One way to perform this is to make systems less complex. For example, whereas a closed-loop system might leave farmers “in the dark” about why particular changes are being made by the system, an open-loop system might give the farmers more control and understanding of what is happening. For instance, in an open-loop system, farmers might be notified that something is wrong or sub-optimal. The farmers could then investigate and decide whether there is indeed a problem and how they want to address it. Because the PLF system would not be making as many decisions on their behalf, farmers would not need to demand as much transparency from it. Similarly, if a PLF system were designed so that the farmers could control more features of how it operated, that could also obviate some of the need for transparency. For example, if farmers could choose what temperatures they wanted the system to maintain, what amount of food to provide, what levels of activity they expected from the animals, and so on, they would be in control over more variables and thus less dependent on receiving assurances that the system would handle these variables in accordance with their preferences. This is a tradeoff with the kinds of efficiencies and emergent animal management practices sometimes promised by PLF but might be worth that loss for those with less trust in the technologies, at least initially.

Another general strategy for addressing transparency challenges is to provide acknowledgements of the major values that guided the development of particular PLF systems. This strategy is an alternative to providing numerous details that could become overwhelming and impractical to disclose. For example, rather than providing extensive details about all the input and output variables associated with a PLF system and the ways those values were analyzed, the developers of a system could focus on the main features of the system (e.g., animal welfare or economic efficiency) that the system was designed to maximize. In order to avoid being misleading, it would also typically be important to clarify how those features were defined or conceptualized (e.g., how animal welfare was measured) and how trade-offs are handled (e.g., what the system does when it would cost more to maximize particular features of animal welfare). Along these lines, the developers could also provide general information about how carefully their systems had been developed and under what range of conditions they had been implemented, which could serve as a proxy for more detailed information about the reliability of the systems. As mentioned above, it is sometimes the case that people might not be aware of the implicit values they hold or are expressing in their work. In such cases, going through guided processes of dialogue to draw these out could be an important step before acknowledging them [6,38].

To address issues of trust and motivated communication, several strategies can be employed. For example, third-party experts who have higher trust from multiple parties and who have different interests can verify claims. This could include regulators, activist watchdogs, the media, or engineers and designers working for a non-profit institution such as a university. Alternately, increasing the community epistemic capacities of both potential users and developers of a PLF system so that they can verify claims, formulate better questions, and communicate their values more clearly can mitigate a lack of trust [39].

Finally, although machine learning algorithms create significant complications for achieving transparency, there are some steps that can be taken to address these challenges. For example, the field of “explainable AI” (XAI) explores ways to clarify some of the features of machine learning algorithms even if they cannot be fully understood [34], such as by running “sensitivity analyses” to determine which input variables make the most difference to the predictions that a system provides. Sometimes uncertainty estimates can also be provided so that the users of a machine learning system have a sense of its reliability. Thus, the use of machine learning algorithms in PLF systems is not a barrier to providing at least some forms of transparency. Additionally, in keeping with the strategy discussed previously of making PLF less complex, machine learning could be added only in a gradual fashion so that its predictions could initially be compared to human judgment in open-loop processes. This comparison could provide its own form of transparency, as users could see the ways in which machine learning complemented, recapitulated, or differed from their own judgment.

## 4. Conclusions

Transparency is an important tool for promoting public engagement with emerging technologies and navigating ethical disagreements or concerns between various stakeholders. However, the subject matter of PLF technologies—animals and food—makes transparency even more important in this context than in many other scientific and technological contexts. People have a (literally) visceral reaction to the food they eat, which makes them wary of novelty. Moreover, PLF is wading into the complex relationship our food system has with non-human animals raised to be food; we acknowledge that they have welfare concerns that we must respect, but we eventually take their lives for the benefit of humans. In addition, producers, distributors, regulators, and all the other actors in the food system must balance desires for adaptation and innovation with a need to preserve the current system’s ability to function. The fraught discourse around genetically modified organisms and other forms of agricultural biotechnology provides a good example of how a lack of meaningful transparency early in research and development can contribute to catastrophic loss of trust; it is an example that PLF developers would do well to learn from and avoid.

## Figures and Tables

**Figure 1 animals-13-03358-f001:**
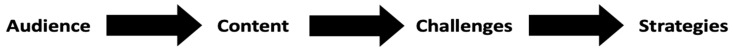
Representation of a framework for achieving transparency in the context of PLF.

**Table 1 animals-13-03358-t001:** A sample of major audiences, content, challenges, and strategies to be considered as part of transparency efforts.

Audience	Content	Challenges	Strategies
Other scientists and engineers Farmers Consumers Industry groups Regulators	Basic operation, strengths, and weaknesses of PLF system Data generated by the system Basic goals of the designers Implicit values of the system Operation and nature of underlying ML algorithms	Difficulty identifying and communicating with relevant audiences Not knowing the information to be disclosed Lack of trust Problematic motivations of the developers or communicators Opaqueness of ML algorithms	Collaborations with end users Design to minimize transparency needs Acknowledgment of major guiding values Independent verification Community epistemic capacities Explainable AI

## Data Availability

Not applicable.

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
