# Peer review of "A Framework for Transparency in Precision Livestock Farming"

_animals, 2023, doi:10.3390/ani13213358_

Round 1

Reviewer 1 Report

Comments and Suggestions for Authors

In this work, Elliott and Werkheiser addressed transparency in precision livestock farming (PLF) by proposing a framework. This is a timely discussion as PLF is an emerging technology. The title of this manuscript implies that this would be a research article or review paper in the field of animal science. However, Sections 3-6 are overwhelmed with opinions of the authors. A balance between scientific soundness and opinions is needed. Further, the authors may improve the article by addressing the specific comments/questions below.

Lines 50-52: most PLF technologies are proof-of-concept i.e., they are applied on small scales which requires frequent supervision and interruption throughout the technology development. Currently, PLF is in its infancy and far away from the intelligent “closed loop” system. The authors need to elaborate on the closed loop system and justify why they are introduced in the context. In addition, the reference here was dated. More recent applications and references are needed.

Lines 71-83: the arguments are valid only given contexts. For instance, the aim of PLF technologies is to improve farming efficiency, and there are applications to specifically assist smallholder farmers, which does not necessarily eliminate agricultural jobs. I would encourage not merely mentioning concerns of PLF technologies but instead a comprehensive review of PLF benefits/concerns considering the ethical and social aspects.

Sections 3-6: The narrative mainly focuses on four pillars: audience, content, challenge, and strategy. The authors presented opinions within each pillar while there lacked global coherence. A concrete example will be helpful to align the proposed framework with real-world issues for PLF transparency.

Section 5 (Challenges): the PLF technology seems not to be the cause that led to some of the challenges mentioned in this section. PLF is usually considered an interdisciplinary field where computer scientists/engineers, animal scientists, and practitioners collaborate. It’s not the transparency issue that made engineers not understand the big picture of PLF systems but their training background or the role they play in the team. For instance, an engineer may develop a perfect computer vision system for PLF while he or she has never been to a farm. On the other hand, for those who want to understand the algorithms behind PLF, it’s not the transparency impeding their understanding but the lack of background knowledge. Transparency is not usually the limiting factor in this case, as most PLF applications, especially in academia, would publish the algorithms.

Taking a step back, even before the PLF's emergence, many of the challenges already existed. These problems, especially in the US, arose due to the challenges of traceability. The authors may further discriminate the challenges of PLF and those from traceability issues. This helps the general readers to understand what the core challenges of PLF could be.

Author Response

We greatly appreciate the time the reviewers took to engage with the manuscript, and we found the comments quite helpful. We have made a number of changes based on the recommendations (discussed below) and improved the manuscript as a result. In what follows, for ease of discussion we have enumerated the comments.

1) In this work, Elliott and Werkheiser addressed transparency in precision livestock farming (PLF) by proposing a framework. This is a timely discussion as PLF is an emerging technology. The title of this manuscript implies that this would be a research article or review paper in the field of animal science. However, Sections 3-6 are overwhelmed with opinions of the authors. A balance between scientific soundness and opinions is needed. Further, the authors may improve the article by addressing the specific comments/questions below.

Glad this was interesting to the reviewer! We would classify the content of the paper not as “opinion” but rather as the use of philosophical methods (e.g., conceptual clarification) that are standard in our field. We suspect there are some differences in disciplinary expectations here. To help address these disciplinary expectations, we have clarified at the end of Section 2 that we are using philosophical methods of analysis to address this issue.

2) Lines 50-52: most PLF technologies are proof-of-concept i.e., they are applied on small scales which requires frequent supervision and interruption throughout the technology development.

We certainly agree that PLF is very much in its infancy. The current sentence on lines 46-47 in the draft the reviewer read states “Some of these tools are currently in use, and many others are in development.”  For clarity we are rewriting this sentence to further emphasize that these technologies are developing.  

3) Currently, PLF is in its infancy and far away from the intelligent “closed loop” system. The authors need to elaborate on the closed loop system and justify why they are introduced in the context.

Closed loop management as mentioned in the cited articles is an example of the kinds of PLF technologies being developed. It is important to show the promises in PLF to readers before discussing the worries users might have and the importance of transparency. To highlight how closed loop management is implicated in the worries discussed in the rest of the paper (e.g., in lines 210, 277, 344), we have added a sentence indicating that these will be discussed more later.

4) In addition, the reference here was dated. More recent applications and references are needed.

We have added two citations on closed-loop PLF from 2023.

5) Lines 71-83: the arguments are valid only given contexts. For instance, the aim of PLF technologies is to improve farming efficiency, and there are applications to specifically assist smallholder farmers, which does not necessarily eliminate agricultural jobs. I would encourage not merely mentioning concerns of PLF technologies but instead a comprehensive review of PLF benefits/concerns considering the ethical and social aspects.

A full comprehensive review of PLF benefits and concerns is beyond the scope of this paper, which presents a framework for transparency which can be applied to many of those concerns. Thus, it is important to mention a few of the concerns to show the reader a) that there are potential problems, and b) that at least some of these problems can be mitigated through transparency following our framework. For clarity, that purpose for mentioning concerns has been restated with a new sentence: “

6) Sections 3-6: The narrative mainly focuses on four pillars: audience, content, challenge, and strategy. The authors presented opinions within each pillar while there lacked global coherence. A concrete example will be helpful to align the proposed framework with real-world issues for PLF transparency.

We appreciate this suggestion, and we agree that a thorough application of the framework to a long case study would be an interesting extension of the paper and one that we would like to see eventually, but we think it is beyond the scope of initially laying out our framework. In lieu of an overarching case study, we have provided examples for each pillar in their respective sections. Rather than providing a single, real-world example to tie them together, Table 1 at the end of those sections lists examples in each of the pillars to show some of the possible combinations of concerns. We would also note that this may be a case where different disciplinary expectations are at work. We would classify the information we provided within each pillar not as “opinions” but as examples of the kinds of issues and concerns that could be captured by the pillars.

7) Section 5 (Challenges): the PLF technology seems not to be the cause that led to some of the challenges mentioned in this section. PLF is usually considered an interdisciplinary field where computer scientists/engineers, animal scientists, and practitioners collaborate. It’s not the transparency issue that made engineers not understand the big picture of PLF systems but their training background or the role they play in the team. For instance, an engineer may develop a perfect computer vision system for PLF while he or she has never been to a farm. On the other hand, for those who want to understand the algorithms behind PLF, it’s not the transparency impeding their understanding but the lack of background knowledge. Transparency is not usually the limiting factor in this case, as most PLF applications, especially in academia, would publish the algorithms.

We partially agree with the reviewer here, but we also think the reviewer’s comment reflects a misunderstanding of our conception of transparency, so we have clarified our views in a couple places in this section. A central point of Elliott’s (2022) original taxonomy of transparency is that transparency is a richer concept than merely publishing algorithms. If a particular audience can’t understand the algorithms, then pursuing transparency for that audience would involving trying to explain the nature or significance of the algorithms in a way they could appreciate based on their background knowledge. Similarly, we think that it would be helpful in many cases to help engineers understand some of the key issues on a farm, and we would classify that as a form of transparency. We have now tried to clarify these points.

8) Taking a step back, even before the PLF's emergence, many of the challenges already existed. These problems, especially in the US, arose due to the challenges of traceability. The authors may further discriminate the challenges of PLF and those from traceability issues. This helps the general readers to understand what the core challenges of PLF could be.

This is a good point and well taken. We have now clarified these points.

Reviewer 2 Report

Comments and Suggestions for Authors

Review: A Framework for Transparency in Livestock Farming

Thank you for inviting me to review this interesting manuscript.  The manuscript reviews an important aspect of precision livestock farming (and other technologies) that is in danger of being over-looked in the race to develop ever-more advanced technology for use on farms. 

Overall, I felt this is well written and provides a good starting point for addressing transparency in precision livestock farming technology that hopefully future researchers might build on and progress.  This also has the potential also to inform policy decisions in the future.  I feel that sections three to six (describing the different aspects of the proposed framework) would benefit from more referencing to support some of the views expressed and improve the clarity as to why the researchers chose to include these aspects into the proposed framework.

I have some minor suggestions for revision (see below).       

Minor comments

General structure: At the end of section two, the framework for achieving transparency is introduced and then subsequent sections each cover individual parts of the proposed framework.  I find this makes it seem like sections three to six are stand-alone sections, whereas they are actually referring to connected parts of the proposed framework.  I suggest that section three is titled ‘proposed framework’ (or similar) and then within section three there are subsections for audience, content, challenges and strategies.  This would emphasise the connection between these sections more obviously.  Your conclusion would then become section four. 

Title: Consider referring to ‘a proposed framework’ to more accurately describe what this manuscript is aiming for  

Line 41: ‘Yick’ in the citation ‘Benjamin & Yick’ is spelt differently to the reference on Line 431.  Please check the spelling of this surname and correct whichever is the incorrect one. 

Line 51-52: Your example (Frost et al.) is 20 years old – consider replacing with a more recent reference as technological advances are rapid and relevance can be lost very quickly  

Line 45: Consider replacing ‘gather’ with ‘collect’ and/or ‘generate’ for a more accurate description.

Lines 45-46: This line does not read well as you start with ‘sensors that can gather data through…’ followed with a mixed list of types or data and types of sensor.  Consider rephrasing this so that either types of data collected, or types of sensor available, are referred to. 

Line 58-59: Are you able to provide a citation to support your statement that ‘as the number of animals on farms increases, it can be more difficult to ensure their welfare’.  This view is often made to support technology use on farm but the correlation between farm size and welfare is weak.  If you are asserting that this is the case, a citation is needed.  Otherwise, consider rephrasing this statement to be less dogmatic. 

Line 88: I think most readers will not be familiar with Blockchain – consider including a brief explanation of what this is for clarity. 

Lines 106-108: This quotation refers to Elliot 2021 and Heald 2006.  This is confusing – only the primary reference should be cited, please correct this.  Please also use quotation marks to clearly indicate this is a direct quote (if it is). 

Lines 170-177: Several assumptions are made in this section about the wants of different audiences, but these are not evidenced.  I would suggest that, either you provide some evidence to support these views (i.e. citations of studies that found for example that consumers in general do not care about the working of the systems), or re-phrase this section to make clear that these are unsupported assumptions using phrases such as ‘it is likely that…’ or ‘ it is possible…’ etc.

Line 193: It is unclear why 'content' is italicised at the end of this line.    

Line 269: Creel et al. does not require a page number here as it is a journal paper. 

Line 407: Consider rephrasing ‘food they put in their bodies’ with ‘food they eat’ for improved accuracy and professional tone. 

Line 460: The first initial of authors should come after the surname, not before.  Please correct the formatting of this reference (Norton et al.)

Line 479: Please capitalise the title of the book appropriately using title case

Line 485-486: This reference is missing a date.  If it is not yet published, please state ‘in press’ or ‘submitted for publication’ or ‘ in peer review’ or ‘unpublished’, as appropriate.  ‘Forthcoming’ is not standard referencing style and does not clearly indicate the status of the reference. 

Table 1: This table is missing a row heading and the formatting needs correcting so that the bullet points align.   

References: Please re-format this manuscript so that references are cited numerically in square brackets as required in the Animals author guidelines.  Please also organise the reference list so that the references are in the order that they are cited in the text, not alphabetical order.  See MDPI style guide for more information. 

Author Response

We greatly appreciate the time the reviewers took to engage with the manuscript, and we found the comments quite helpful. We have made a number of changes based on the recommendations (discussed below) and improved the manuscript as a result. In what follows, for ease of discussion we enumerated the comments. Our responses are added in-line below each numbered comment. 

1) Thank you for inviting me to review this interesting manuscript.  The manuscript reviews an important aspect of precision livestock farming (and other technologies) that is in danger of being over-looked in the race to develop ever-more advanced technology for use on farms.

We are quite glad the reviewer liked the manuscript! We agree of course that there is a real danger in overlooking transparency. 

2) Overall, I felt this is well written and provides a good starting point for addressing transparency in precision livestock farming technology that hopefully future researchers might build on and progress.  This also has the potential also to inform policy decisions in the future.  I feel that sections three to six (describing the different aspects of the proposed framework) would benefit from more referencing to support some of the views expressed and improve the clarity as to why the researchers chose to include these aspects into the proposed framework.

We have added more references to the text.

I have some minor suggestions for revision (see below).       

Minor comments

3) General structure: At the end of section two, the framework for achieving transparency is introduced and then subsequent sections each cover individual parts of the proposed framework.  I find this makes it seem like sections three to six are stand-alone sections, whereas they are actually referring to connected parts of the proposed framework.  I suggest that section three is titled ‘proposed framework’ (or similar) and then within section three there are subsections for audience, content, challenges and strategies.  This would emphasise the connection between these sections more obviously.  Your conclusion would then become section four. 

This is a very helpful recommendation for clarity, so we have made this change and re-organized sections 3-6 into subsections.

4) Title: Consider referring to ‘a proposed framework’ to more accurately describe what this manuscript is aiming for  

We certainly agree that it is a proposal, but our inclination is that using the indefinite article “a” at the beginning of the title makes that clear enough. (Also, we have clarified that the paper is using philosophical methods to propose a framework, which may also alleviate this reviewer’s concerns here.) If the reviewer thinks strongly about this, though, we are willing to reconsider.

5) Line 41: ‘Yick’ in the citation ‘Benjamin & Yick’ is spelt differently to the reference on Line 431.  Please check the spelling of this surname and correct whichever is the incorrect one. 

We have resolved this typo.

6) Line 51-52: Your example (Frost et al.) is 20 years old – consider replacing with a more recent reference as technological advances are rapid and relevance can be lost very quickly  

We have added two citations from 2023.

7) Line 45: Consider replacing ‘gather’ with ‘collect’ and/or ‘generate’ for a more accurate description.

We have changed “gather” to “generate”

8) Lines 45-46: This line does not read well as you start with ‘sensors that can gather data through…’ followed with a mixed list of types or data and types of sensor.  Consider rephrasing this so that either types of data collected, or types of sensor available, are referred to. 

We have added the phrase “and analytic tools” to cover both kinds of data.

9) Line 58-59: Are you able to provide a citation to support your statement that ‘as the number of animals on farms increases, it can be more difficult to ensure their welfare’.  This view is often made to support technology use on farm but the correlation between farm size and welfare is weak.  If you are asserting that this is the case, a citation is needed.  Otherwise, consider rephrasing this statement to be less dogmatic. 

We have added a citation

10) Line 88: I think most readers will not be familiar with Blockchain – consider including a brief explanation of what this is for clarity.

We have provided a short description and added a citation. 

11) Lines 106-108: This quotation refers to Elliot 2021 and Heald 2006.  This is confusing – only the primary reference should be cited, please correct this.  Please also use quotation marks to clearly indicate this is a direct quote (if it is). 

This is a quotation from Elliott 2021, and the citation to Heald was at the end of the quotation. We agree that it is clearer to just cite Elliott as the source of the quotation, and we have placed it in quotation marks.

12) Lines 170-177: Several assumptions are made in this section about the wants of different audiences, but these are not evidenced.  I would suggest that, either you provide some evidence to support these views (i.e. citations of studies that found for example that consumers in general do not care about the working of the systems), or re-phrase this section to make clear that these are unsupported assumptions using phrases such as ‘it is likely that…’ or ‘ it is possible…’ etc.

We have now made sure that we say it is “likely” or “probable” that each audience would want particular sorts of information, and we have added some citations to support our claims. In addition, we had already stated at the end of this section that it is important to work with the audiences to find their actual needs, but this has now been made clearer.

13) Line 193: It is unclear why 'content' is italicised at the end of this line.    

It was italicized for emphasis, but the italicization has been removed for clarity.

14) Line 269: Creel et al. does not require a page number here as it is a journal paper.

Typo fixed. 

15) Line 407: Consider rephrasing ‘food they put in their bodies’ with ‘food they eat’ for improved accuracy and professional tone. 

Rewritten for clarity

16) Line 460: The first initial of authors should come after the surname, not before.  Please correct the formatting of this reference (Norton et al.)

Typo fixed.

17) Line 479: Please capitalise the title of the book appropriately using title case

Typo fixed

18) Line 485-486: This reference is missing a date.  If it is not yet published, please state ‘in press’ or ‘submitted for publication’ or ‘ in peer review’ or ‘unpublished’, as appropriate.  ‘Forthcoming’ is not standard referencing style and does not clearly indicate the status of the reference. 

Fixed.

19) Table 1: This table is missing a row heading and the formatting needs correcting so that the bullet points align. 

Fixed  

20) References: Please re-format this manuscript so that references are cited numerically in square brackets as required in the Animals author guidelines.  Please also organise the reference list so that the references are in the order that they are cited in the text, not alphabetical order.  See MDPI style guide for more information. 

Fixed

Reviewer 3 Report

Comments and Suggestions for Authors

The paper is relevant to the journal/ conference. It looks like more of an opinion paper/ mini review. It can be improved by clearly defining the objectives and justifications for the selection of this topic. Further, authors need to declare that this is their own opinion and not representing their institutions.  

Author Response

We greatly appreciate the time the reviewers took to engage with the manuscript, and we found the comments quite helpful. We have made a number of changes based on the recommendations (discussed below) and improved the manuscript as a result. 

  • The paper is relevant to the journal/ conference. It looks like more of an opinion paper/ mini review. It can be improved by clearly defining the objectives and justifications for the selection of this topic. Further, authors need to declare that this is their own opinion and not representing their institutions.
    • We think that one of our revisions (namely, clarifying that this paper uses philosophical methods of analysis) helps to clarify our objectives, and the revised first paragraph of Section 2 clarifies our justification. We think it is generally understood that those who employ philosophical methods are reporting their own judgment/analyses and not the views of their institutions, but if the editors would like us to clarify that in, say, our declaration of conflicts of interest, we could do so.

Round 2

Reviewer 1 Report

Comments and Suggestions for Authors

All comments were properly addressed by the authors, and my concerns about the last version of the manuscript were cleared. I wanted to especially thank the authors for the response in 7).